# Molecular Mechanisms during Hepatitis B Infection and the Effects of the Virus Variability

**DOI:** 10.3390/v13061167

**Published:** 2021-06-18

**Authors:** Marina Campos-Valdez, Hugo C. Monroy-Ramírez, Juan Armendáriz-Borunda, Laura V. Sánchez-Orozco

**Affiliations:** 1Centro Universitario de Ciencias de la Salud, Departamento de Biología Molecular y Genómica, Instituto de Biología Molecular en Medicina, Universidad de Guadalajara, Guadalajara 44340, Jalisco, México; campos.ibt@gmail.com (M.C.-V.); hugo.monroyram@academicos.udg.mx (H.C.M.-R.); armdbo@gmail.com (J.A.-B.); 2Escuela de Medicina y Ciencias de la Salud, Tecnológico de Monterrey, Campus Guadalajara, Zapopan 45201, Jalisco, México

**Keywords:** acute hepatitis B, chronic hepatitis B, immune pathogenesis, acute liver failure, HBV sensing, HBV variability, HBV interference

## Abstract

The immunopathogenesis and molecular mechanisms involved during a hepatitis B virus (HBV) infection have made the approaches for research complex, especially concerning the patients’ responses in the course of the early acute stage. The study of molecular bases involved in the viral clearance or persistence of the infection is complicated due to the difficulty to detect patients at the most adequate points of the disease, especially in the time lapse between the onset of the infection and the viral emergence. Despite this, there is valuable data obtained from animal and in vitro models, which have helped to clarify some aspects of the early immune response against HBV infection. The diversity of the HBV (genotypes and variants) has been proven to be associated not only with the development and outcome of the disease but also with the response to treatments. That is why factors involved in the virus evolution need to be considered while studying hepatitis B infection. This review brings together some of the published data to try to explain the immunological and molecular mechanisms involved in the different stages of the infection, clinical outcomes, viral persistence, and the impact of the variants of HBV in these processes.

## 1. Introduction

HBV is a small hepatotropic DNA virus where an envelope is formed by a lipid bilayer with the small, middle, and large size surface antigens (S-HBsAg, M-HBsAg, and L-HBsAg) embedded as transmembrane proteins [1,2]. The envelope surrounds a nucleocapsid—formed by 120 core proteins (HBcAg) dimers—that stores the viral genome and polymerase (HBVPol) [3,4] (Figure 1a).

The viral genome consists of a 3.2 kb relaxed circular DNA (rcDNA) molecule. It contains four open reading frames (ORFs) encoding for polymerase (P), surface (S), pre-core/core (C), and X (Figure 1b and Figure 2) [5]. HBV replication and transcription are controlled by its regulatory elements, such as enhancers, promoters (S1, S2, pre-core, core, and X); and signals of encapsidation (ε), polyadenylation, and replication (direct repeats 1 and 2, DR1 and DR2) [4].

An important trait of the viral genome is the overlap of its ORFs; around 52% of the nucleotides code for more than one viral protein. Deletions and non-synonymous mutations in overlapping regions impact the HBV genotypes evolution and the differentiation of the pre-S domain (key in the infection process) [6]. The evolution rate of HBV has been estimated at around 1.4 × 10^−4^–5.7 × 10^−5^ substitutions per site per year [7], being the pre-S/S region the most heterogeneous of HBV genome [8].

The variability of the HBV genome is driven not only by viral fitness and the lack of corrective activity by the HBVPol, but mutants also result from their selection through pressure exerted by endogenous (host’s immune pressure) and exogenous (antiviral therapy and vaccination) factors [8,9]. It has been found that there is a strong selection in the pre-S1 region, which overlaps with nucleotides encoding the spacer domain of the HBVPol, which is relatively adaptable to non-synonymous mutations. The accumulation of these variants provides flexibility for conformational structural changes and responds to immune selective pressure [10,11].

## 2. HBV Replicative Cycle

The virus replicative cycle (Figure 3) starts with binding L-HBsAg with the sodium taurocholate contransporting polypeptide (NTCP) to mediate the entry of the viral particle into hepatocytes [12]. The virus is endocytosed and transported to the nucleus while the viral capsid is de-enveloped [13,14]. The viral genome seems to enter the nucleus through the nuclear pores and is repaired by host factors and transformed into the form of covalently closed circular DNA (cccDNA). This replication intermediate associates with host proteins to form a viral minichromosome that functions as a template for viral RNA transcription [15,16,17]. Five viral mRNAs are produced (Figure 2), including the pregenomic RNA (pgRNA), which is the main intermediate in the viral replication. pgRNA is exported to the cytosol where its 5′-end binds to the HBVPol and induces the packaging process [4,18]. Then, inside the nucleocapsid, the pgRNA is reverse transcribed by the viral polymerase into a new-rcDNA. Alternatively, a double-strand linear DNA (dslDNA) is formed and plays an important role in the HBV-DNA integration into the host genome [4,18]. The mature capsids can either deliver the rcDNA into the nucleus, increasing the cccDNA content or binding to HBV surface proteins in the endoplasmic reticulum (ER) to be secreted as virions outside the hepatocytes [16].

HBV variants could emerge during the replicative cycle by different mechanisms, such as (a) cytosine deamination induced by host APOBEC (apolipoprotein B mRNA editing enzyme) enzymes, (b) errors incorporated during pgRNA transcription, (c) failure in the host DNA repair mechanisms, (d) error-prone activity of the HBVPol (during DNA polymerization dependent of either RNA or DNA), (e) generation of different HBV forms (such as HBV dslDNA and spliced forms that can produce novel fusion proteins), and (f) viral recombination [19].

## 3. HBV Genotypes and Variants

HBV has been classified into nine genotypes (A–I), one putative genotype (J), and at least 35 subtypes (Table 1) [5]. The virus genotypes and subgenotypes are differentiated by >7.5% and 4–7.5% differences in all nucleotides of the viral genome, respectively. Subgenotypes are subdivided into clades which differ by <4% [20]. Moreover, based on the HBsAg heterogeneity, nine serological subtypes (ayw1-ayw4, ayr, adw2, adw4, adwq, adr, and adrq–) have been identified. Moreover, there is a significant correlation between serological subtypes and genotypes: adw being associated with genotypes A, B, F, G, and H, adr with C, and ayw with D and E; however, many exceptions have been reported [21].

Many studies describe the frequency of HBV mutations; a good example is a recent large-scale analysis of HBV genome sequences (*n* = 6479, genotypes A–H) in which they report that immune escape mutations were present in 10.7% of the sequences (genotype B showing the highest rate). They also found that hepatocellular carcinoma (HCC) associated mutations were present in 33.7% of the sequences and that the overall frequencies of lamivudine-, telbivudine-, adefovir-, and entecavir-resistant mutants were 7.3%, 7.2%, 0.5%, and 0.2%, respectively, while only 0.05% showed reduced susceptibility to tenofovir [9].

In 2020, Velkov et al. published the results of an HBV nucleotide sequences analysis of the global occurrence of clinically relevant virus variants. They identified differences in the clinical manifestation of HBV infection between genotypes and their geographical locations [23].

### 3.1. Clinical Relevance of HBV Diversity during Infection

There is scientific information that compiles, summarizes, and analyzes what is known about HBV genotypes (epidemiology, clinical relevance, variants, and geographical distribution) [21,24,25,26]. Some of the most relevant data are:(a)The association of genotype A with durable remission after HBeAg seroconversion. Those patients infected with genotypes A and B have higher rates of HBsAg seroclearance than genotypes D and C, respectively [27].(b)Infection with genotype A is an independent risk factor for progression to chronic hepatitis B (CHB), and the treatment with nucleoside/nucleotide analogues (NAs) does not prevent progression to chronicity [28].(c)Genotype C is more prone to cause CHB compared to genotype B [29].(d)Genotype D appears to be more prevalent in patients with HBV-related acute liver failure [30].(e)Genotype C and D are associated with a higher viral load compared to genotypes B and A, respectively. Infection with genotype H is associated with a low viral load; nevertheless, it is difficult to elucidate their association with liver diseases since patients infected with this genotype usually have additional risk factors, such as alcohol consumption, co-infection with hepatitis C virus, and obesity [31,32].

### 3.2. Drug-Resistant Variants

NAs inhibit the activity of HBVPol by targeting its RT domain. The major downside of this therapy is the apparition of drug-resistant mutations due to long-term treatment. The development of drug resistance is caused by primary HBVPol (RT domain) mutations, which affect NAs recognition and binding, and therefore allowing HBV replication but with a reduced fitness compared to the wild-type HBV strain. These primary mutations usually go along with compensatory mutations that restore and improve the HBV fitness affected by primary mutations [23,33]. The major mutational patterns in NAs resistance are the RT domain mutations rtL180M/rtM204(I/V) (for lamivudine, entecavir, telbivudine, and clevudine) and rtA181V/rtN236T (for adefovir and tenofovir (TDF)) [34]. In a large-scale analysis, rtM204V/I and rtL180M were identified as the most abundant mutations known to cause resistance. Furthermore, mutation rtA194T (one of the few tenofovir resistance mutations [35]) was identified in several genotypes, albeit in very low numbers, except for genotype H [23]. 

It was recently observed that differential response to tenofovir (HBV rapid clearance or on-drug persistence) might be associated with the genetic variations across the whole HBV genome and may be specific for each genotype. Analysis of the inferred 3D structures showed no difference in the TDF recognition and binding by the RT-domain of HBVPol among intra-host HBV variants that rapidly decline or persist in the presence of TDF [36]. It is interesting that the mutation rtA194T in combination with HBeAg negative and pre-core or basal core promoter (BCP) mutants increase the replication capacity of the drug-resistant mutants and thereby restores the viral fitness in vitro. This would indicate that patients with HBeAg negative chronic infection may be at risk of developing drug resistance to TDF [35]. 

HBVPol RT domain region is affected by its overlap with the *S* gene. A lower rate of synonymous substitution within this overlapping region has been reported; the most conserved RT regions were located within the YMDD motif and the N-terminal that are critical for RT functionality. These regions correspond to highly conserved HBsAg domains which are crucial for its secretion [37]. An analysis of the amino acid (aa) substitution rate in the overlapping region of the *S* gene and the RT domain in patients with CHB reported that those with detectable aa substitution in either/both of RT/S sequences had significantly lower serum HBV-DNA and HBsAg levels compared to patients without detectable RT/S aa substitutions [38]. Another consequence of the RT/S overlap is that some drug-resistant mutations may even escape vaccine-induced anti-HBsAg antibodies [39].

The presence of resistant-to-NAs mutations has been extensively studied due to the importance of these types of drugs for HBV infection treatment. These mutants in naïve-treatment patients should be considered during the natural history of the infection for the patients’ effective treatment selection.

### 3.3. HBV-Related HCC

Risk factors associated with HCC development over time are advanced age, male sex, elevated alanine aminotransferase (ALT) level, positive HBeAg, and high HBV-DNA levels [40]. In addition, numerous studies support the hypothesis that HBV genotypes diversity and variants may influence the risk of HCC development.

A report from India shows that a prevalence of 50% for genotype D and 25% for genotypes A and C each affects HBV-related HCC cases [41]. Before this, it was described that genotype D and mixed genotypes (A + D) infections have a higher risk of developing HCC compared to genotype A [42]. Moreover, subgenotype D1 might have a greater oncogenic potential compared to subgenotype D3 [43].

In Taiwanese and Japanese patients, the risk of HCC was higher in genotype C compared to genotype B and A [40,44,45,46]. In addition, BCP double mutation T1762/A1764 is more frequent in genotype C compared to genotype B increasing the risk for HCC [47]. In those patients infected with genotype B, non-cirrhotic HCC early onset is more common; whereas, genotype C is associated with cirrhotic HCC late-onset [48]. Nonetheless, chronically HBV-infected Japanese patients treated with NAs indicated that those infected with genotype B had a higher probability of HBsAg loss during treatment than those infected with genotype C [46]. While in Asia, genotypes B and C have been compared as the most prevalent in that region, a study in Alaska, where different genotypes are prevalent, a significant association between genotype F infection with the development of HCC was found, independent of the presence or absence of BCP mutations [49]. Besides BCP and pre-core mutations, viral variants within the X and pre-S1 region are also associated with the risk of HCC development. Failure in NAs treatment associated with HBV mutations allows the replicative state of HBV raising the viral load and, together with the previously mentioned mutations, increases the risk of HCC development [48]. Teng et al. demonstrated that the presence of deletions into the pre-S region is related to the risk or recurrence of HCC [50]. In Japan, a study performed with 40 HBV genomes (95% genotype C) obtained from HBsAg-positive HCC patients noticed that deletions and miss-sense mutations were frequent in the pre-S2 region. Furthermore, about 90% of the isolated HBV had double T1762/A1764 BCP mutation, which highlighted its relevance in HCC development [51]. Excessive HBsAg accumulation in the ER could be inducing the activation of unfolded protein response (UPR) and leading to ground glass hepatocytes (GGHs) [52]. GGHs have been reported as HBV-related HCC markers and represent precursor lesions. It has been demonstrated that pre-S deleted proteins may promote HCC development by activating oncogenic signaling pathways in hepatocytes [53,54,55,56,57]. Different types of GGHs contain specific pre-S mutations, type I harboring pre-S1 deletions and type II containing deletions over pre-S2 [58]. The ER stress induced by pre-S2 deletions causes DNA damage, centrosome overduplication, and genomic instability, and the presence of these mutants in sera frequently develops resistance to NAs [59]. Type II GGHs are more likely to be in cases of HCC and are associated with advanced fibrosis in CHB [60].

## 4. Natural History of the HBV Infection

In highly endemic areas, perinatal and horizontal transmission (especially from an infected to an uninfected child during the first 5 years of life) are the most common routes [61]. Infants born to mothers HBsAg and HBeAg positive are at a higher risk of acquiring HBV infection (transmission risk: 70–100% in Asia and 40% in Africa) than those born to HBsAg positive and HBeAg negative (5–30% in Asia and 5% in Africa) [62]. HBV horizontal transmission is especially important in children because of the high risk of acquiring chronic asymptomatic infection. Children with CHB are more prone than adults to be HBeAg positive and to have a high viral load. Significant amounts of HBV-DNA have been found in body fluids (tears, saliva, sweat, and urine) of CHB carriers, including children [63,64,65,66,67]. Due to children’s frequent close person-to-person contact between them, they are at a higher risk of infection [68]. In low-prevalence areas, HBV spreads through sexual contact and percutaneous (primarily injecting drug use) route; in this setting, the infection usually manifests as acute hepatitis [69]. At least 95% of immunocompetent adults with symptomatic acute hepatitis B (AHB) will recover spontaneously [70]. In total, 80–90% of those infected by vertical transmission develop CHB, while 30–50% and <5% among those infected before being 6 years old and during adulthood, respectively, evolve to chronicity [61,70,71,72].

### 4.1. Acute Hepatitis B

The incubation period of AHB ranges from one to four months post-infection. Symptomatic AHB is primarily an adulthood disease, while two-thirds of the patients remain asymptomatic. The clinical presentation of HBV infection can be a serum-sickness-like illness characterized by fever, arthralgias, and rash, which may occur in the prodromal period. It is followed by constitutional symptoms, anorexia, nausea, jaundice, and right upper quadrant discomfort. Serum biochemical markers are characterized by elevated ALT activity and usually a mild elevation in bilirubin [69].

Acute liver failure (ALF) is the most serious complication of AHB; however, it happens in <0.5% of patients. The ones at risk of this outcome are those infected with BCP and/or pre-core mutations, those coinfected with other hepatitis viruses, and/or those with underlying liver disease [69].

The resolution of infection is accompanied by anti-HBs seroconversion ≥10 IU/mL and viral clearance. Nonetheless, there are cases of hidden infection distinguished by the persistence of cccDNA in the nucleus of infected hepatocytes. The persistence of HBV in recovered patients even many years after the onset of the infection has been proved by: (a) the persistence of HBV-DNA and HBV-specific CD8^+^ cytotoxic T lymphocytes (CTLs), (b) the reactivation of HBV with HBsAg seroreversion while receiving immunosuppressive therapy, and (c) infections by liver transplantation to HBV non-immune recipients from donors with inactive infection [73,74,75,76,77,78].

Estimates indicate that virus reactivation can occur in approximately 20–30% of chronic carriers [79]. In carriers with inactive infection or with evidence of resolved hepatitis B, reactivation can occur and manifest itself by: (a) an active process of necroinflammation with a ≥5-fold increase in serum ALT activity above the normal upper limit, (b) absolute increase on ALT levels ≥100 IU/mL associated to ≥1 log_10_ elevation of viral load compared to the pre-exacerbation period, or (c) absolute HBV-DNA increase exceeding 10^8^ copies/mL regardless of ALT activity levels [80]. The occurrence of reactivation often causes a flare of disease that can be severe and result in ALF; nevertheless, it could also be transient and clinically silent. Reactivations may resolve spontaneously in most instances, but if immune suppression continues, the re-establishment of chronic hepatitis occurs and can lead to progressive liver injury and cirrhosis [81].

### 4.2. Chronic Hepatitis B

The establishment of chronic infection is the result of the synergy between host and viral factors, and it occurs when viral clearance is ineffective. It is defined by the persistence of serum HBsAg for at least 6 months [61,70]. 

In 2017, the European Association for the Study of the Liver (EASL) proposed a new nomenclature for CHB stages. Its foundation is the two main characteristics of chronicity: infection vs. hepatitis [82]. The stages shown in Table 2 are defined by viral, biochemical, and fibrosis markers for liver disease; they are not necessarily sequential [73,82].

## 5. Immunopathogenesis of HBV Infection

### 5.1. Immune Response to HBV in the Evolution of the Infection

HBV is a non-cytopathic virus, and the basis for liver disease seems to be immune-mediated with CTLs inducing the death of infected hepatocytes [84,85]. During the progress of pathogenesis, different immune components are engaged, having interferon-γ (IFN-γ) and CTLs concerted action as the main protagonists. In general, during AHB, viral infection is detected and restrained by a strong and robust immune response accompanied by viral clearance, whereas, in patients with CHB, both innate and adaptive immune responses are weak and rarely achieve viral clearance [86].

### 5.2. Response in AHB Infection

The main obstacle of studying the immune response during early AHB in humans is the difficulty of detecting and recruiting patients in the early stages of hepatitis B infection since most patients present the highest viremia before the onset of clinical manifestations and by the time of the typical presentation of jaundice, the maximal reduction in circulating HBV has already occurred. That is why the first evidence of the role of the immune response during the incubation period of the HBV (the time between infection and the onset of symptoms) came from animal models. Nevertheless, there are some relevant longitudinal studies performed in patients that follow from the clinically silent incubation to the recovery or clearance of HBV. A disadvantage is that this evidence came mainly from blood markers since performing a liver biopsy in patients with acute hepatitis is not recommended and involves ethical issues. 

From these studies, the evidence suggests that in the early stages of acute infection, innate and adaptative responses play a role in which inhibition of viral replication and clearance of viral DNA could be mediated by natural killer (NK) cells, CD56^+^ natural T cells (NT), and CD4^+^ and CD8^+^ T cells (Figure 4) [87,88,89]. In patients with asymptomatic AHB with a transient low-grade viremia, NT cells producing IFN-γ were detected before the highest peak of viremia, suggesting a non-cytolytic response involved in the initial viral control. It was followed by the maximum NK cell-cytotoxicity and IFN-γ production at the time of the highest viremia [87]. At this point, CD4^+^ and CD8^+^-mediated responses were still very weak, and NK/NT cells and their capacity for IFN-γ production is dropped off quickly; these effects are correlated with an increase in IL-10 levels [87,88,89]. In a transgenic mice model, it was demonstrated that CTLs collaborate during the acute phase of infection by releasing tumor necrosis factor α (TNF-α) and INF-γ without destruction of HBsAg positive hepatocytes [90]. These cytokines inhibit HBV replication and reduce cccDNA through the activation of nuclear apolipoprotein B mRNA editing enzyme, catalytic polypeptide-like 3G (APOBEC3) deaminase, which is responsible for HBV cccDNA degradation [91]. The viral DNA reduction performed mainly by INF-γ and TNF-α precedes the liver damage, which is characterized by elevations of ALT levels in serum and T cells liver infiltration [86,89,92,93,94].

Another cell population involved is CD1d-restricted NKT cells; these cells are activated and have a role in the innate response by causing acute hepatitis and producing IFN-γ and IL-4 in transgenic mice expressing viral antigens [96]. Along with cells of the immune system, hepatocytes produce interferon-α (IFN-α) and interferon-β (IFN-β), which inhibits viral packaging [97].

Early NK cells activation and capacity for IFN-γ production were reduced during the viremia peak, and this reduction was correlated with increased IL-10 levels. Data has indicated that in patients with symptomatic AHB type I interferons (IFNs-I), IL-15, and IFN-γ are not appropriately induced in response to the infection, but they found that in the early stages of AHB, there is induction of IL-10, and its surge accompanies the temporary inhibition of NK cells (at the peak of viremia) and T cell responses [88]. 

After the NK/NT cells decay in patients with AHB, the adaptative T cell response handled by CD4^+^ and CD8^+^ T cells increases progressively, reaching and sustaining (mainly by the CD8^+^ T cells subset) maximal IFN-γ production when viremia is already undetectable. Interleukin 12 (IL-12) secretion by T cells reaches a peak before IFN-γ, while interleukin 4 (IL-4) and IL-10 production by CD4^+^ cells gradually increase, and the greatest production is observed when the secretion of Th1 cytokines (IFN-γ, IL-2, and TNF-α) are already declining, and the anti-HBs antibody titers are increasing in serum [87,88,89]. 

In symptomatic AHB patients after viremia control, high levels of ALT were detected during 3–9 weeks [88,89]. The presence of HBsAg, HBeAg, and the increase in ALT and aspartate aminotransferase (AST) in serum characterized the acute infection [97]. In patients with resolved infection and self-limited swollen liver, viral clearance was performed by a strong immune response (vigorous, broad, and polyclonal) by specific T cells against HBV antigens [84,95,98]. After the clearance of viral antigens and the increase in ALT and AST, the appearance of anti-HBc followed by anti-HBe and anti-HBs antibodies take place. Anti-HBc (initially both IgM and IgG) appears one to two weeks after HBsAg secretion. IgG persists during chronic infection, and IgM-anti-HBc can be present in some patients with severe exacerbations of chronic HBV infection, though the title is lower than that during acute infection [84,86,92,97,98,99].

To counteract the liver damage resulting from the vigorous immune response, regulatory T (Treg) cells play an important role. In a murine model of acute HBV infection, an increase in the number of Treg cells was detected soon after the onset of HBV infection. These cells mitigate the immune-mediated liver damage through down-regulation of effector T cells, and as a consequence, the antiviral activity is hampered. However, these Treg cells do not influence the development of HBV-specific CTLs and memory T cells. Moreover, they seem to control the recruitment of innate immune cells to the liver [100]. These observations could support the idea that Treg cells increase at the beginning of the infection and favors non-cytotoxic mechanisms in this stage, delaying the liver damage and extending the virus clearance. 

Studies in chimpanzees have shown that intrahepatic virus levels and histological evidence of inflammation are correlated with serum levels of HBV-DNA and ALT activity. CTLs mediate cytopathic mechanisms for the elimination of infected hepatocytes. The repertoire of CD8^+^ epitopes during acute infection is diverse for HBVPol, HBsAg, HBxAg, and HBcAg. Responses to HBx and HBVPol were observed before viremia, followed by a reduction during viremia, and then CD8^+^ positive cells were recovered responding to HBVPol, HBsAg, and HBcAg. CD4^+^ T cells followed the same pattern of epitope recognition but were most responsive to HBcAg [84,86,92,97,98]. In the absence of human biopsies, these pieces of evidence can be correlated with intrahepatic viral control and liver damage during the incubation period. All this data together could lead to ponder that the initial innate response against HBV is dominated by a non-cytolytic response mechanism mediated by NK cells, with minor participation of immune adaptative T cell response. Subsequently, liver damage markers are accompanied by the CD8^+^ T cells’ cytolytic effector activity. This evidence is supported by a model of acute woodchuck hepatitis virus infection [101].

### 5.3. Response in CHB Infection

During CHB, HBV persistence is due to the accumulation of cccDNA in infected cells leading to the viral RNA transcription keeping up [95,102]. At this stage, due to the long-lasting exposure to viral antigens, T cells become exhausted and functionally impaired. Exhausted HBV-specific CTLs highly express inhibitory receptors (such as programmed cell death protein 1 (PD1); T cell immunoglobulin domain and mucin domain-3 (TIM3); cytotoxic T-lymphocyte antigen 4 (CTLA4) and 2B4) [95,103,104,105,106,107,108]. In addition, HBV-specific T cells are susceptible to apoptosis because of the increased expression of Bcl2-interacting mediator (BIM) or TNF-related apoptosis-inducing ligand (TRAIL) receptor 2 (TRAILR2), which is targeted by TRAIL^+^ liver NK cells. Added to this, the number of CD4^+^CD25^+^FOXP3^+^ Treg cells increases in blood and liver and suppresses HBV-specific effector T cells (Figure 5) [95,105,109,110,111,112].

Although TRAIL expression is increased and the cytotoxic activity of NK cells is not impaired during chronic HBV infection, their activation and subsequent IFN-γ and TNF-α production are strongly hampered. This functional alteration of NK cells is caused by immunosuppressive cytokines, such as IL-10 and transforming growth factor-β (TGF-β), which are mainly produced by Kupffer cells (KC) and hepatic stellate cells (HSC), respectively, as well as by immune cells [113,114,115].

Dendrite cells (DCs) isolated from CHB patients and in vitro matured myeloid DCs (mDCs) have also decreased their allostimulatory capacity. The percentage of these cells, which express CD80 and CD86, is decreased as well as their capacity to produce TNF-α in response to stimuli. Similarly, isolated plasmacytoid DC (pDCs) from CHB patients produce fewer IFN-α in response to stimuli [116]. In addition to this response, it has been seen that circulating arginase I and granulocytic myeloid-derived suppressor cells (gMDSC), which are also accumulated in the liver, increase in HBV-replicating phases without immunopathology, and inhibits T cells by depriving them of L-arginine [117].

Studies in murine models that were generated by crossing female hemizygous HBV transgenic mice with male non-transgenic mice of the same genetic background resulted in HBV negative mouse pups (TGD mice). These mice were used to study the mechanism of HBV persistence after vertical transmission (hydrodynamic injection to introduce 1.3 mer HBV genomic DNA into mouse hepatocytes). The results showed that KCs mediate the impairment of CTLs’ response, favoring HBV persistence in these mice. Maternal HBeAg could play an important role in up-regulating the inhibitory ligand PD-L1 on KCs; as a consequence, TGD mice macrophages could be sensitized by maternal HBeAg since this antigen could be able to cross the mouse placenta or it could be delivered during colostrums’ feeding. Binding PD-1 and PD-L1 between CD8^+^ T cells and hepatic macrophages were responsible for CTLs exhaustion in offspring of HBV-infected TGD mice [118]. Another study performed in a murine model of HBV persistence-induced systemic tolerance supports the role of IFN-γ as an inductor of tolerance. HBV antigens persistence induced sustained IFN-γ secretion by hepatic (T helper) Th cells; this cytokine promotes CXCL9 secretion from KCs and supports the retention of antiviral CD4^+^ T cells in the liver and their eventual apoptotic elimination partially via CTLA4 ligation [119]. IFN-γ may have a dual role during HBV infection; on one hand, it is known its relevant role in the spontaneous elimination of HBV at the onset of the infection; on the other hand, IFN-γ contributes to maintaining liver tolerance. The specific T cell differentiation pattern defined by the cytokines surrounding the liver tissue could influence the IFN-γ role.

In HBV-infected patients, expression of IL-22 is increased compared to healthy subjects. IL-22 seems to induce migration of intrahepatic Th17 cells and promotes the progression of liver fibrosis in CHB patients [120]. Furthermore, patients with cirrhosis associated with hepatitis B with more severe disease have an increase in Th17 cells and IL-17 in plasma [121]. Both interleukins could play a role in promoting HSCs activation secreting several chemokines to promote Th17 cell chemotaxis [120,121]. 

Longitudinal and transversal studies performed in patients with CHB and animal models with viral persistence make clear that a weak adaptative immune response, which could emerge as a consequence of an inadequate innate response, is the most related cause of chronicity. This immune failure allows a sustained HBV replication, transcription, and translation of its genome products. These products lead to an increase in the HBV-DNA, RNA, and viral proteins, causing qualitative and quantitative changes in hepatic and blood cytokines, which impair the immune system and favor the viral persistence and mutation. Even with all the scientific progress on this topic, it is necessary to increase research, especially in geographical regions where only a few studies are giving scarce information on the HBV genotypes that are prevalent in those areas.

## 6. Acute Liver Failure in HBV Infection

HBV-associated acute liver failure (ALF) (also known as fulminant hepatitis B) is a dramatic clinical syndrome ending with a fatal outcome or liver transplantation in most cases. ALF occurs when the regenerative capacity of the liver is overcome by the rate and extent of hepatocellular death. ALF may occur after acute HBV infection in a person without previous liver disease (primary infection) or during an acute exacerbation of CHB, leading to multiorgan failure. ACLF can occur spontaneously or as a result of immunosuppression due to chemotherapeutic or immunosuppressive agents [122].

### 6.1. Liver Failure in Acute Infection

ALF is characterized by the presence of coagulopathy and hepatic encephalopathy in the absence of prior hepatic disease. ALF may be the result of an exaggerated immune response. Noteworthy, patients that survive without liver transplantation will seroconvert their HBsAg status (HBsAg negative/anti-HBs antibodies positive) and will not develop chronic hepatitis [123].

Coexisting factors, such as alcoholic hepatitis and co-infection with hepatitis delta [124], are associated with HBV-ALF. In addition, infection with HBV harboring double BCP mutation (A1762T/G1764A), pre-core mutations G1986A, G1899A, and A2339G, predominantly HBeAg negative, genotype B1/Bj, and higher levels of HBV-DNA is common in ALF compared to patients with acute self-limited hepatitis B [125]. 

To investigate the molecular bases of liver damage on the site of HBV replication, interesting research has been performed using liver samples from patients with HBV-associated ALF and chimpanzees with classical acute hepatitis for comparison. All patients with ALF were infected with HBV/genotype D. In addition to pre-core stop codon mutation G1896A that was invariably present in ALF, they also identified two to three amino acid changes in the B cell epitope (aa 73–85). These mutations were associated with increased HBcAg expression ex vivo and were absent in chimpanzees and patients with acute hepatitis B. Moreover, liver gene and miRNA expression profiles in ALF patients showed an up-regulated intrahepatic B cell gene signature promoting IgM assembly and secretion, targeting HBcAg with nanomolar or even picomolar affinities, followed by the complement cascade activation with the subsequent massive liver necrosis [126].

### 6.2. Acute-on-Chronic Hepatitis B Liver Failure

Patients with a poor outcome of HBV-related acute-on-chronic liver failure (ACLF) had an increase in Treg cells, and these cells were redistributed from periphery to liver. Besides, these cells were located mainly in areas with an inflammatory profile or with a lymphocyte-rich profile, supporting their role in the liver damage and inflammation since this cell pattern was not observed in normal liver. Additionally, there was a positive correlation con HBV-DNA loads and circulating Treg cells [127]. Additional studies to understand the effect of Treg cells and ACLF must be performed; nonetheless, it is possible that high levels of peripheral Treg cells in non-survivors do not restrain the liver inflammation due to severe uncontrolled immune modulation. 

Serum levels of IL-1β, IL-6, IL-8, IL-10, and TNF-α in patients with ACLF were significantly higher compared to the normal control group. Besides, it was found that IL-10 increased before the time of diagnosis, and this increase was positively correlated with the ALT level [128]. In a comparative study performed in patients with ACLF who eventually died, patients with ACLF who survived, and patients with mild–moderate CHB, the IL-1, IL-10, and TNF-α levels were significantly higher in the group of patients who died. These cytokines have an important role in the antiviral activity; nevertheless, their increase with the concomitant failure in viral clearance in patients with ACLF who die might reflect the degree of liver inflammation and damage. Additionally, IL-12 in the non-survival group was higher than in the survival group at the diagnosis stage, indicating an intensive reaction that tries to eliminate the viral infection [129]. Nonetheless, there are contradictory results about the IL-10 levels in CHB patients [130,131,132,133]; further studies with a huge number of patients must be performed. In CHB patients with ALF, the levels of TGF-β1 and IL-31 (a member of the IL-6 family) are significantly increased with a positive correlation with total bilirubin (Tbil) and α-fetoprotein (AFP). In those patients who recovered from liver injury, TGF-β1 and IL-31 levels decreased; and in non-survivors, they were markedly up-regulated. The TGF-β1/IL-31 pathway has been associated with disease severity in HBV-related liver cirrhosis [134] and could play an important role in the pathogenesis of liver injury in ACLF [135].

Additionally, in the previously mentioned reports, a Th17/Treg axis imbalance that could have a role in the development of ALF in CHB has been observed. Nevertheless, as well as the studies related to IL-10, the results are diverse; in some studies, it was found a higher frequency of Th17 cells in ACLF, whereas others did not find significant changes in Treg cells [136,137,138,139]. 

### 6.3. *Viral Factors in ACLF*

Most of the studies that analyze viral markers in HBV-ACLF patients have been made in Asia, where genotypes B and C predominate. HBV genotype B has been associated with the risk of ACLF. Even so, it is important to highlight that an associated risk with the HBV genotype in ACLF was not found in a meta-analysis [140]. Some potential identified triggers of ACLF are the presence of mutations in the pre-core/core region, mainly the pre-core G1896A mutation and BCP A1762T/G1764A double mutation [141,142,143]. In addition to the previous mutation referred, a more frequent mutation index in Pre-S, HBcAg (aa 90–135 in genotype C and aa 60–130 genotype B), and HBxAg (aa 131–135) regions were identified in ACLF patients compared to acute hepatitis, chronic hepatitis, and immune tolerant HBV carriers by sequencing a total of 606 HBV full-length genomes (from 49 treatment-naïve HBV-infected patients) [144]. The mechanism with the presence of these mutations and their clinical significance needs to be further investigated. It is to mention that information related to other HBV genotypes is scarce at this clinical stage; therefore, it is a must to increase these kinds of studies in geographical areas where different genotypes from B and C are prevalent.

## 7. Sensing and Response to HBV

The host cells involved in the recognition of HBV are hepatocytes, hepatic non-parenchymal cells, and innate immune cells [145].

Hepatocytes: Primary human hepatocytes (PHHs) and differentiated HepaRG (dHepaRG) cells express pattern recognition receptors (PRRs) such as retinoic acid-inducible gene-I (RIGI), melanoma differentiation-associated protein 5 (MDA5), and most Toll-like receptors (TLRs). Both RIGI and MDA5 are important for sensing viral RNAs during viral infections [146]. After the recognition of viral RNAs, these receptors induce the stimulation of an IFN-β promoter, increasing the synthesis of this antiviral cytokine. The loss of MDA5 in Huh7 cells and in MDA5 knockout mice caused an increase in HBV replication; however, the overexpression of MDA5 did not impact IFN-β induction. In that same study, it was found that RIGI did not inhibit viral replication [147]. In contrast, the recognition of the epsilon region of the pgRNA by RIGI and how this recognition not only induced the production of type III IFNs but also counteracted the interaction of HBVPol with the pgRNA [148]. Infection of PHHs cells and dHepaRG with HBV induced a weak and transient innate response (production IL-6, IL-29 and type I IFNs) [149]. Another in vitro model of micropatterned cocultures of PHHs with stromal cells (MPCCs) incubated with HBV-infected serum stimulated the expression of interferon-stimulated genes (ISGs) (Figure 6) [150].

The innate nuclear sensor interferon-inducible protein 16 (IFI16) can recognize the cccDNA and even play an antiviral role by mediating epigenetic regulation of the viral chromatin. On the other hand, it was found that IFI16 is downregulated in HBV-infected hepatocytes and in CHB patients [151].

It was recently demonstrated in monocyte-derived dendritic cells (MDDCs) that HBV-RNAs (including mRNAs and pgRNA) are not immunostimulatory, but naked HBV-DNA can be sensed by the cyclic GMP-AMP synthase/stimulator of interferon genes (cGAS/STING) pathway (Figure 6). Again, it was noticed that this pathway is expressed at low levels in PHH, being unable to respond to productive HBV infection, but it is able to sense naked HBV-DNA when it is present in sufficient amounts [152].

Hepatic non-parenchymal cells: Liver sinusoidal endothelial cells (LSECs) play a key role in the uptake of viral particles circulating in the blood to infect adjacent hepatocytes. LSECs seem to be unable to replicate the virus, but they may serve as a reservoir of endogenous reinfection. They produce large amounts of anti-inflammatory cytokines (such as TGF-β) and constitutively express major histocompatibility complex-I restricted antigens and co-stimulatory molecules that could favor the shift of the hepatic immune balance towards tolerance [153,154].

In vitro assays have proved that KCs (even while being uninfected) are able to sense the HBV and activate nuclear factor kappa B (NF-κB); subsequently, these cells release IL-6 and other pro-inflammatory cytokines (IL-8, TNF-α, and IL-1β) transiently; although, there is no induction of IFN response. IL-6 activates the mitogen-activated protein kinases exogenous signal-regulated kinase 1/2 and c-jun N-terminal kinase, which inhibits hepatocyte nuclear factor (HNF) 1α and HNF 4α expression, two essential transcription factors for HBV gene expression and replication [155].

Dendritic cells: The exposure of BDCA1^+^ mDCs to HBsAg results in their strong maturation, cytokine production, and an enhanced capacity to activate antigen-specific CTLs. It was also found that CD14 and TLR4 play a crucial role in the HBsAg-mediated DCs maturation [156].

## 8. Viral Epitopes

Many HBV variants that present mutations within B and/or T cell epitopes have been identified. An example is the rtI169M mutation that also affects the T cell epitope located from 157 to 162 aa in the HBsAg [157]. In addition, multiple mutations in the core gene, as well as the G1896 pre-core mutation, were identified in patients during the immune clearance phase or in the low replicative phase, while in patients during the immune tolerant phase, only minimal changes were identified. These findings support that evolution of new HBV variants may be induced mainly by the host immune pressure rather than viral reverse transcription error rate [158].

The emergence of HBV mutants causes the existence of heterogeneity (genetic and antigenic) among the virus variants throughout the population and even in an individual. A systemic review from 2008 showed that there is a significant variation of T-cell epitopes between the HBV genotypes [159].

### 8.1. HBsAg

The HBsAg contains four conserved trans-membrane (TM) regions consisting of an α-helix structure that maintains the stability of the protein structure. The targets of T cell and B cell epitopes in the HBsAg are concentrated in the hydrophilic loops between the α-helices that harbor more variable aa residues. That includes the major hydrophilic region (MHR, aa 99–160), which contains a conformational B cell epitope cluster. In addition, the core part of MHR denominated as “a” determinant (aa 121–147) harbors a cluster of epitopes targeted by neutralizing anti-HBs antibodies [10,160,161]. Escape mutants’ emergence within the “a” determinant (induced or not by vaccination) and how they can affect the host immune response has been previously reported by several studies [162,163,164]. The presence of these mutations in the B cell epitopes has been associated with conformational changes that may lead to vaccine escape, diagnosis failure, and immune tolerance [157,161,165,166,167].

### 8.2. Immune Escape Mutations and Diagnosis Failure

HBsAg is a major viral protein in the induction of protective immune response. The detection of this protein in serum is the main diagnostic tool of the infected individuals. The most common HBV mutants associated with immune escape are those with changes in aa 145. Furthermore, there are mutations in the *S* gene that result in occult HBV infection (presence of HBV-DNA in the absence of HBsAg) [168,169]. 

A large-scale analysis (*n* = 6434), found that immune escape mutations were present in 10.7% of the sequences (I/T126S, 1.8%; G145R, 1.2%; M133T, 1.2%; and Q129R, 1.0%), being G145R the first vaccine escape mutant identified and I/T126S the most frequent [9]. These mutations have been previously related to vaccine failure [170,171]. These studies have identified aa 126 (I126S/N and T126A, 29.63%) and aa 145 (G145R/A, 25.93%) as the hottest mutations post immunization period [9,170,171].

A study performed in Argentina that analyzed HBsAg and HBVPol sequences from 530 samples of HBV-infected individuals found that diagnostic failure mutants (DFM) and vaccine escape mutants (VEM) were detected in 10.7% and 7.5%, respectively. The most frequent DFM was Y100C in subgenotype A1 and was frequently detected in HBsAg negative samples [172]. Despite that, even if this mutation has been frequently identified in the lack of HBsAg detection in serum samples, in vitro assays suggested that this substitution alone is not enough to evade HBsAg detection by commercial ELISA assay [173].

In 2020, it was found that the overall frequency of escape mutations is low and evenly distributed among genotypes; only P127H/L mutations (associated with occult infection) appear to be the wild-type amino acid for genotypes E, F, and H (96.8%, 98.9%, and 97.7%) [23,157].

Another interesting observation is that HBV reactivation in patients with resolved infection seems to be triggered by an immunosuppressive status, characterized by the expansion of variants associated with immune escape and with MHC class II-restricted T cell epitope variants (L21S and/or F220C variants in the S-HBsAg) and/or impaired virion secretion (E2G, L77R, L98V, T118K, and Q129H in the S region, and M1I/V in the pre-S2 region) [174].

### 8.3. HBcAg and HBeAg

Mutations at pre-core can affect the structure of the HBeAg and even eliminate its expression. For example, the mutation G1896A creates a stop codon that not only avoids the expression of the HBeAg but also strengthens the secondary folding structure of the encapsulation signal on the viral pgRNA and leads to an increase in viral replication. This mutation affects the immunological response because the presence of HBeAg normally stimulates regulatory T cells that suppress T CD8^+^ cells against HBcAg. This mechanism evokes an unregulated immune response against infected hepatocytes, lowers the viral load, and diminishes the infectious state. HBeAg negative infections are frequently characterized by higher nucleotide diversity compared to HBeAg positive infections. This behavior is probably explained by the increase in the replication rate, combined with an increasing selection pressure in HBcAg [7].

The major immunodominant regions of the HBcAg have been mapped to residues 78–83. Around the most protruding HBcAg region (aa 71–87), there are the major B cell epitopes to HBV mainly between aa 76 and 82; another B cell epitope lies around aa 129–132. [175,176].

The characterization from 1 to 150 aa sequence in HBcAg strains from patients infected with genotype D found that mutations in HBcAg were more frequent in those strains from asymptomatic carriers than HBeAg negative patients with CHB. Additionally, HBV-DNA serum was inversely associated with the presence of mutations in CTLs epitopes of HBcAg, while it was directly associated with the presence of double promoter T1762/A1764 together with G1757 mutations. This inverse correlation of HBV-DNA level and CTL escape HBcAg mutations in HBeAg seroconverted patients with CHB favors CTL escape mutations selection. This correlation also establishes the persistence of HBV infection despite viral fitness reduction [176]. 

HBcAg peptide from aa 81 to 95 elicits IFN-γ by CD8^+^ T cells, and from aa 131 to 145 induces IFN-γ and IL-2 production by CD8^+^ and CD4^+^ T cells in favor of HBV clearance [87]. 

## 9. The Role of TLRs

Toll-like receptors (TLRs) might have a real role in the clearance of HBV [97]. There are various studies of the interaction between TLRs and HBV, but there is no convincing evidence of HBV proteins, viral RNAs, and HBV-DNA being truly recognized by TLRs. Regardless of the fact that TLRs mediated innate immunity is not, or only weakly activated by HBV infection, experimental activation of the TLR system was able to suppress HBV replication in vitro and in vivo [177].

Specific ligands for TLR3, TLR4, TLR5, TLR7, and TLR9 had the ability to control HBV replication in HBV transgenic mice in an IFN dependent manner. In the HepG2 cell line, ligands of TLR2, TLR3, TLR4, TLR7, and TLR9 were reported to suppress HBV replication. Besides, in this cell line, the overexpression of TIR-domain-containing adapter-inducing interferon-β (TRIF), IFN-β promoter stimulator 1 (IPS), as well as myeloid differentiation primary response 88 (MyD88) was found, which may be involved in the suppression of HBV replication (Figure 6) [145,178].

## 10. Interferon Response

HBV is usually referred to as a stealth virus; however, some studies have shown that the virus can be sensed by immune cells and induces IFNs response. It has been demonstrated by in vivo and in vitro models that the production of type I INFs (IFN-α and IFN-β) is poorly induced during HBV infection [148,179,180]. Even though HBV does not trigger type I IFN production, IFN-α and -β can suppress HBV replication in vitro and in HBV transgenic mice models. Recombinant IFN-α has been approved and successfully used as a standard treatment for chronic HBV infection [178]. Some of the ISGs induced by IFN and their role against HBV are reviewed by Pei and Lu, 2014 [178].

A genome-wide association study that tried to identify genetic variants associated with early sustained response to peg-IFN in CHB patients did not find any significant hits but found that *G3BP2 rs3821977* was associated with peg-IFN response in HBeAg negative patients. *G3BP2* has a role in the IFN pathway. *G3BP2* G-allele differentially affected interferon-inducible protein 10 (IP-10), IL-10, and IL-8 protein expression [181].

### HBV Genotypes in the Response to IFN

HBV genotype is a predictor of IFN treatment. Among HBeAg-positive patients, genotype A infection is associated with a higher rate of HBeAg and HBsAg loss. Studies in Asia have shown that genotype B is associated with a higher rate of HBeAg loss compared to genotype C. Similar findings have been demonstrated among HBeAg negative patients, with better response rates in patients with genotype A versus genotype D [182].

However, it was observed in cell culture-based HBV infection models that the sensitivity of HBV to IFN-α in hepatocytes may be determined more by the cell-intrinsic IFN response than by the viral genotype [183].

An IFN-stimulated response element (ISRE)/IFN regulatory element (IRE) has been identified in the EnhI/X gene promoter region of the HBV genome (nt 1091-1100) [184]. It has been observed that a single base mutation on this region can affect IFN response; for example, a single change in the fourth base (C➔T) may partially decrease the effect of IFN [185]. This base change could be the reason why genotype B has a relatively higher response to IFN-α therapy compared to genotype C [186].

The expression of spliced variants—HBV splice-generated protein, a major spliced HBV-RNAs-encoded proteins (truncated TP domain) and N-terminal-truncated viral polymerase protein (common in most of the known spliced HBV variants)—result in strong suppression of IFN-α signaling transduction in CHB patients) [187].

## 11. HBV Interference against the Antiviral Activity

Only around 30% of CHB patients respond well to treatment with exogenous IFN-α [188,189]. There is evidence that HBV has mechanisms to counterattack type I IFNs signaling routes. Even though it has been seen that HBV can slightly induce interferon production in some models, it has been proven that the virus also performs strategies to go against those defense mechanisms of the cell. For example, the expression of TLR2, TLR3, TLR4, TLR7, and TLR9 in peripheral blood mononuclear cells (PBMCs) was reduced in chronically HBV-infected patients; this reduction could be the result of the exhaustion of the TLR system under continuous activation. 

In hepatitis B patients, pDCs frequency is decreased (inversely correlated with ALT levels and viral load), and TLR9 expression is reduced, as well as in PBMCs and pDCs from the HBV group, the production of IFN-α is impaired (inversely correlated with serum viral load of HBV) [190,191]. Internalized HBV particles inhibit TLR9-mediated secretion of IFN-α and down-regulating TLR9 transcriptional activity in pDCs and B cells. These changes can interfere with TLR9 activity by blocking the MyD88-IRAK4 axis and Sendai-virus-targeting IRF7 to block IFN-α production. In HBV genotypes A–H, neutralizing CpG motif sequences that have a suppressive effect on TLR9-immune activation have been identified, and they could reduce IFN-α secretion [192].

In PBMCs from HBV-infected patients, TLR-mediated cytokine expression (IL-6 and IL-10) and TLR3-induced interferon expression is higher compared to healthy controls. In addition, TLR3 mediated IFN-γ is inhibited in the presence of serum that contains HBV. In the presence of HBV, KCs and LSECs are stimulated by TLR3 ligands; consequently, IFN-γ, interferon sensitive genes, and proinflammatory cytokines are suppressed, as well as the activation of T cells through TLR3-stimulation [193].

A TLR3-diminished expression in CHB patients has been reported, but it could be restored after entecavir treatment and at a greater recovery degree with peg-IFN treatment [194]. Reduced expression of TLR3 was also found in patients with active chronic hepatitis B compared to asymptomatic carriers; however, in contrast to Huang’s study, they did not find a significant difference in comparison with healthy controls [195].

HBsAg, HBeAg, or HBV nullify TLR-induced antiviral activity by suppressing IFN-β production and subsequent interferon-stimulated gene induction, as well as suppressing interferon regulatory factor 3 (IRF3) activation, NF-κB, and extracellular signal-regulated kinase (ERK) 1/2. In HBV-Met cells infected with HBV, TLR stimulation does not induce antiviral cytokines in contrast to primary hepatocytes. The expression of proinflammatory cytokines (TNF-α and IL-6) stimulated by TLR and activation of IRF-3 is suppressed after up-regulation of HBV replication in HBV-Met cells [196].

PHHs cells and dHepaRG cells infected with HBV induced a weak and transient innate response (production IL-6, IL-29, and type I IFNs). It was also observed that TLR3 and RIGI/MDA5-mediated innate response was inhibited by factor(s) in the HBV inoculums’ (but not being HBsAg or HBeAg) [149].

### Interference by Viral Proteins

(a)HBsAg

HBsAg inhibits the IFR7 expression and nuclear translocation, which results in the suppression of IFN-α production mediated by TLR9 [197]. A correlation between the level of HBsAg in plasma and the impairment in the cytokines production was identified after a challenge to TLR2 and TLR4 ligands on PBMCs from CHB patients, along with observations of lower expressions of TLR1, TLR2, TLR4, and TLR6 on these cells [198].

(b)HBVPol

The viral polymerase interferes with TLR3 and RIGI signaling by blocking the activation of interferon regulatory factors 3 and 7 (IRF3 and IRF7). HBVPol interacts with the DDX3 DEAD-box RNA, which normally is in charge of activating TANK-binding kinase 1 (TBK1) and IκB kinase-ε (IKKε) [199].

STING is a central factor for foreign DNA recognition and antiviral innate immunity. HBVPol inhibits STING-stimulated IRF3 activation by binding HBVPol RT and RNaseH domains with STING, leading to an extreme decrease in STING’s K6-linked polyubiquitination, which is associated with a loss of its functions [200]. Additionally, HBVPol avoids nuclear translocations of signal transducer and activator of transcription 1/2 (STAT1/2) via competitive bidding to importin-α5 and suppresses STAT1 Ser727 phosphorylation by inhibiting protein kinase C delta type (PKC-δ) activation, which together impairs IFN-α induced STAT activation (Figure 6) [201]. 

(c)HBX

The X protein of the HBV downregulates the expression of mitochondrial antiviral signaling (MAVS), which is an important virus-activated signaling pathway that activates NF-κB and IRF3 (Figure 6) [202].

## 12. Conclusions

The scientific advances on hepatitis B since the discovery of HBsAg in 1965 have been evidenced by the improvement of serologic and molecular diagnosis, in addition to the development of effective recombinant vaccines and the antiviral treatments that already control viral replication. However, even if vaccines and antiviral treatments are widespread in the world, low-income and middle-income countries are in trouble to prevent or treat this infection due to the high cost involved in vaccine coverage or for an efficient long-term treatment. Nonetheless, current therapy is not a real cure for the infection, and that is why researchers keep trying to come up with new alternatives for treatment that lead to a cure, such as HBV inhibitors entry as NTPC inhibitors. Moreover, the most significant challenge in the HBV cure is to achieve cccDNA eradication and to block HBV genome integration into the human genome.

Most of the knowledge about HBV genetic variability and clinical implications on hepatitis B comes from longitudinal and transversal studies performed in countries where hepatitis B has been a serious health problem. Those countries have been classified as regions with a high or intermediate prevalence of HBsAg, with an elevated risk to develop chronic hepatitis and end-stage liver diseases where the most frequent genotypes are A, B, C, and D. Nevertheless, there is scarce information about the genotypes F, G, E and H, which are endemic at low or medium-income countries where the disease might be sub-diagnosed.

Hepatitis B clinical presentations and evolution varies depending on the host’s genetic background and HBV variability. That is the reason why we considered reviewing the most relevant aspects of the molecular bases during a hepatitis B infection from clearance to chronic liver disease, including HBV molecular aspects. Therefore, relevant data reviewed has shown us that during the early stages of acute HBV infection, the initial innate response seems to be mainly dictated by the non-cytolytic activity of NK cells followed by cytolytic T cells response accompanied by liver damage. Treg cells play an important role in HBV non-cytolytic clearance reducing the liver damage induced by CTLs. Virus clearance is correlated with a wide and strong immune response; meanwhile, the development of chronic liver disease is correlated with a weak antiviral immune response, as previously mentioned. In addition to the genetic background of the infected host, there are questions that might be resolved in relation to immune response and viral factors, such as (a) what is the role of HBV variability in the clearance or persistence of the infection? (b) Why do some patients with HBV BCP and/or pre-core mutations develop HCC, and other patients carrying the same HBV mutations are asymptomatic or just have mild liver disease? For example, patients infected with genotype H and these mutations are at risk to develop HCC? (c) Is genotype H really associated with mild liver disease? No data are focused on this topic, possibly because most of the patients infected with genotype H are living in low-income areas from Mexico, and it is possible that they die before HCC development due to liver cirrhosis complications. It must be relevant to perform longitudinal studies in countries where genotypes F, G, E and H are prevalent in order to increase the knowledge about their association with the natural history of hepatitis B. Moreover, research must go on since HBV emergence mutations are continuously changing by the immune or treatment pressure and the virus’s low capacity to repair its DNA mistakes. These deficiencies could be overcome by taking more interest in the molecular diagnosis of the infection and improving the strategies for the follow-up of patients. Hence, the obtained results would impact translational medicine.

Chronic hepatitis B stages can be well identified by biochemical and viral markers (e.g., ALT, AST, viral load, etc.), while NK and T cells seem to suffer changes in their activity during the disease progression, according to the disease stages (changes in their phenotypes and production of cytokines). These changes could be generated by the induced tolerant state caused by the expression of inhibitory receptors and the secretion of regulatory cytokines such as IL-10. In addition, TGF-β, IL-22, and IL-17 cytokines are correlated with liver damage progression. It is possible that in the near future, these deregulated molecules could be used as liquid biopsy improving the earlier diagnosis of HBV infection or identifying the chronic stages of liver disease in order to detect more accurate predictors of treatment response.

A relevant issue of interest in the overall research on chronic hepatitis B is the “strategies” employed by the virus to evade and block the antiviral response. These evasion mechanisms, such as diminishing the expression of TLRs and blocking of sensors and cellular signaling involved in the viral response, favor the infection persistence. Research must continue trying to identify efficient cccDNA inhibitors in order to reduce the HBV replication, transcription and antigens synthesis restoring the immune response with an efficient viral clearance.

## Figures and Tables

**Figure 1 viruses-13-01167-f001:**
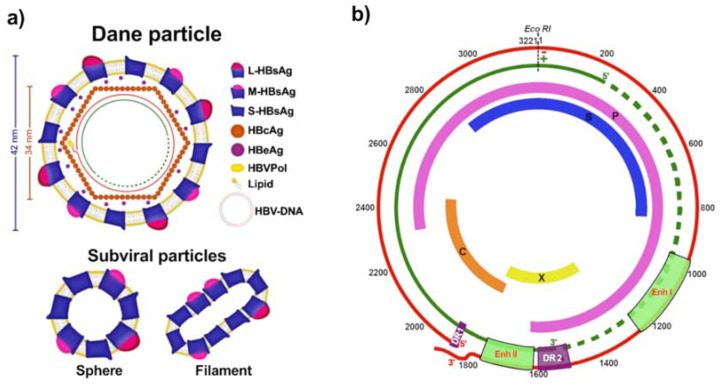
HBV (**a**) particle structures and (**b**) genome and ORFs. In red and green, the negative and positive strands (respectively) of HBV genome; colored boxes indicate the approximate location of the ORFs; green rectangles indicate enhancer regions and purple rectangles direct repeat regions. Abbreviations: HBV, hepatitis B virus; DR, direct repeat; Enh, enhancer; P, polymerase; S, surface; C, core; X, *X* protein gene.

**Figure 2 viruses-13-01167-f002:**
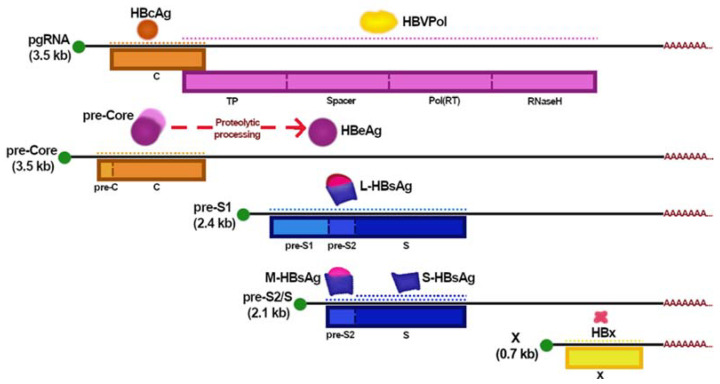
A representation of the HBV mRNAs and the translation of viral proteins. 5′-cap is shown as a green dot and polyA tail in red; the rectangles in colors represent the ORFs while the dotted lines represent the translated segments to produce the viral proteins. Abbreviations: kb, kilobase; pgRNA, pregenomic RNA; HBcAg, HBV core antigen; HBVPol, HBV polymerase; C, core; TP, terminal protein; Pol(RT), polymerase reverse transcriptase; HBeAg, HBV e antigen; S-, M-, and L-HBsAg; HBV small, middle, and large size surface antigens; HBX, HBV X protein.

**Figure 3 viruses-13-01167-f003:**
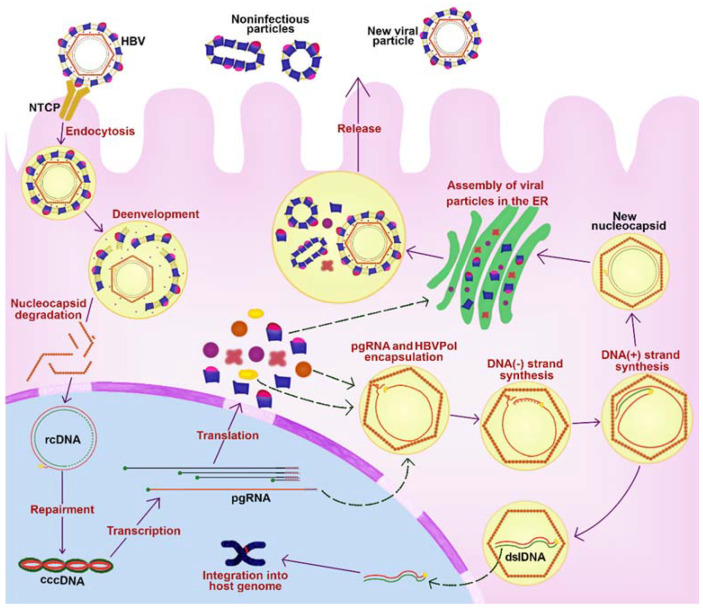
HBV replicative cycle. HBV undergoes endocytosis through the NTCP. Inside the cell, the rcDNA enters the nucleus, where it is repaired to form the cccDNA. The latter forms a viral minichromosome for the transcription of viral RNAs. The pgRNA binds to HBVPol in the cytosol to induce the packaging process that is followed by the synthesis of the HBV-DNA. The new viral particle is assembled in the ER and translocated to be secreted outside the hepatocytes as virions. Abbreviations: HBV, hepatitis B virus; NTCP, sodium taurocholate cotransporting polypeptide; rcDNA, relaxed circular DNA; cccDNA, covalently closed circular DNA; pgRNA, pregenomic RNA; ER, endoplasmic reticulum; HBVPol, HBV polymerase; dslDNA, double-stranded linear DNA.

**Figure 4 viruses-13-01167-f004:**
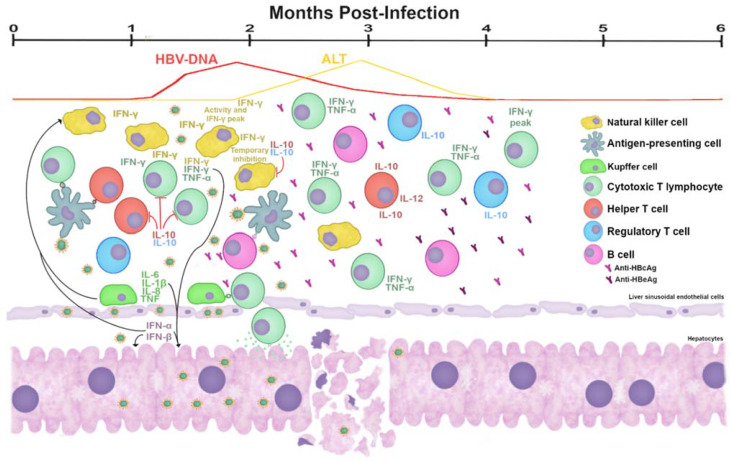
The potential dynamic of the immune response during AHB. Before the peak of HBV-DNA, the antiviral process is primarily mediated by non-cytolytic mechanisms, mainly by NK cells with IFN-γ and TNF-α production. During this stage, the functions of T cells may be slowed down by regulatory cytokines. Infected hepatocytes collaborate by producing IFN-α and IFN-β to fight the infection. KC cells could be participating by stimulating NK cells and by acting as antigen-presenting cells. After the peak of viral DNA in serum, the antiviral activity is performed mostly by T cells producing cytokines and antibodies (anti-HBc followed by anti-HBe). The cytolytic mechanism induced by HBV-specific CTLs leads to serum ALT increase. Serum HBV-DNA and ALT graphic was modified [95]. Cells and their secreted cytokines are in the same color. Trunked red lines indicate inhibitory effects. Abbreviations: HBV, hepatitis B virus; HBV-DNA, HBV DNA in serum; ALT, alanine aminotransferase; NK, natural killer; KC, Kupffer cell; IFN, interferon; IL, interleukin; TNF, tumor necrosis factor; CTL, cytotoxic T lymphocyte.

**Figure 5 viruses-13-01167-f005:**
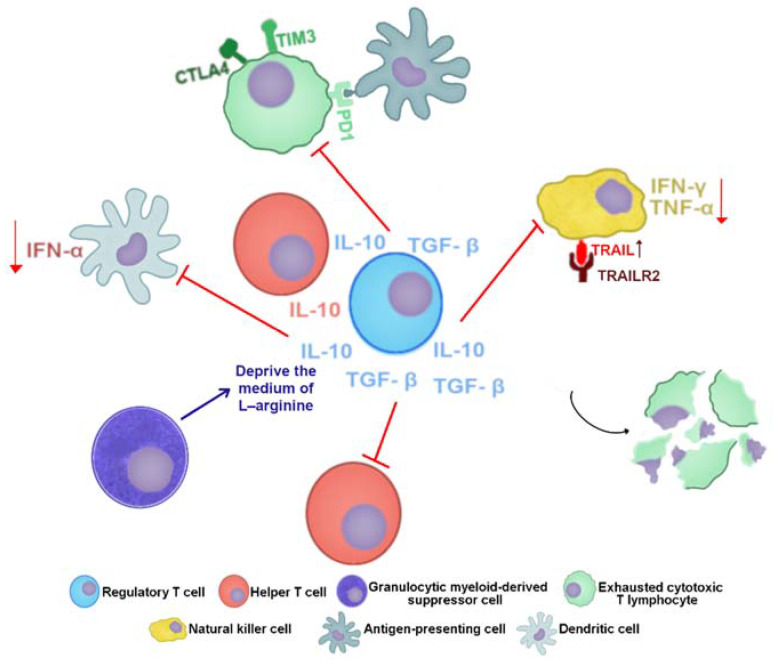
The induction of tolerant state during chronic hepatitis B. T cells are induced to an exhausted state; the secretion of IL-10 and TGF-β by Treg cells inhibits the activity of immune cells such as DCs and NK cells. CTLs express higher levels of TRAILR2, which makes them susceptible to die by interacting with cells such as NK cells, which express higher levels of TRAIL in CHB patients. There is also metabolic regulation conducted by gMDSCs by consuming the L-arginine available in the medium. Cells and their secreted cytokines are colored the same. Trunked red lines indicate inhibitory effects, and red arrows indicate a reduction of secretion. Abbreviations: IFN, interferon; IL, interleukin; TGF-β, transforming growth factor-β; TNF, tumor necrosis factor; TRAILR2; or TNF-related apoptosis-inducing ligand (TRAIL) receptor 2; CTLA4, cytotoxic T-lymphocyte antigen 4; TIM3, T cell immunoglobulin domain and mucin domain-3; PD1, programmed cell death protein 1.

**Figure 6 viruses-13-01167-f006:**
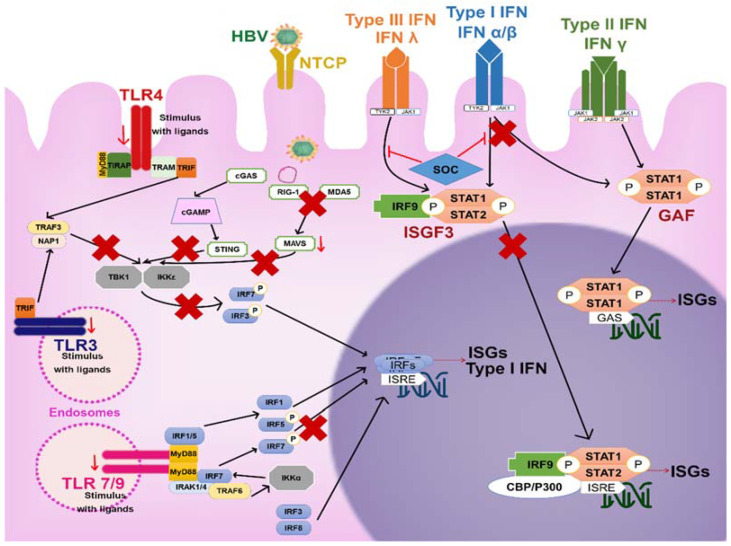
Probable signaling in the mechanism induced by TLRs, IFNRs, RIG-I, and STING in the control of the viral infection and how the virus interferes with them (marked with a red X). The stimulus of TLRs can induce a response to eliminate HBV. HBV presence leads to downregulation of the expression of TLRs (observed in PBMCs; indicated with red arrows) and/or interfere with its signaling (blocking the MyD88-IRAK4 axis, suppressing IRF3 activation, inhibiting the expression and nuclear translocation of IFR7). The virus can also inhibit RIG-I/MDA5-mediated response and STING-stimulated IRF3 activation; blocks: the activation of interferon IRF3 and IRF7, TBK1/IKKε activation, STING-stimulated IRF3 activation and, down-regulate the expression of MAVS. The presence of the virus can also interfere with IFN receptors signaling by avoiding nuclear translocations of STAT1/2 and IFN-α-induced STAT activation. Abbreviations: HBV, hepatitis B virus; NTCP, Na+-Taurocholate co-transporting polypeptide; TLR4, Toll-like receptor 4; TLR7/9, Toll-like receptor 7 and 9; MyD88, Myeloid differentiation primary response 88; TIRAP, TIR domain-containing adaptor protein; TRAM, TRIF-related adaptor molecule; TRAF, TNF Receptor Associated Factor; NAP1, NF-kB–activating kinase-associated protein 1; IRF, interferon regulatory factor; TBK1, TANK binding kinase 1; IKKε, IκB kinase-ε; STING, stimulator of interferon genes; STAT, signal transducers and activators of transcription; IRF9, IFN regulatory factor 9; ISG, interferon stimulated genes; ISGF3, ISG factor 3; SOC; IFN-induced suppressor of cytokine signaling; GAS gamma IFN activated sequence (GAS) gamma IFN activated sequence GAS; MAVS, mitochondrial antiviral signaling; cGAMP, cyclic-GMP-AMP; cGAS, cyclic-GMP-AMP synthase; GAF, IFN-γ activated factor.

**Table 1 viruses-13-01167-t001:** HBV genotypes and their distinguishing features.

Genotype	Genome Size (bp)	Subgenotypes	Distinguishing Features
A	3 221	A1, A2, A4, and quasi-subgenotype A3	6 bp insertion at the 3′-end of core gene (Insertion of aa 153 and 154 in HBcAg) Unusual G1896A mutation Common BCP mutations
B	3 215	B1, B2, B4–B6, and quasi-subgenotype B3	B1 and B5 are pure HBV/B strains B2, B3, and B4 are recombinants with genotype C in the core region
C	3 215	C1-C16	Common BCP mutations
D	3 182	D1-D6	33 bp deletion at the 5′-end of pre-S1 region (Deletion of aa 1-11 in pre-S1)
E	3 212	-	3 bp deletion at 5′-end of the pre-S1 region (Deletion of aa 11 in pre-S1)
F	3 215	F1-F4	Unusual G1896A mutation
G	3 248	-	36 bp insertion at position 190 in the core ORF (Insertion of 12 aa in HBcAg) 3 bp deletion at the 5′-end in the pre-S1 region (Deletion of aa 11 in pre-S1) Stop codons at positions 2 and 28 (G1896A) of the pre-core ORF render it unable to express HBeAg Usually found in coinfection with other genotypes that express HBeAg
H	3 215	-	Unusual G1896A mutation
I	3 210	I1 and I2	Evolved as a recombinant of genotypes A, C, and G
J	3 182		Single HBV isolate identified in an elderly Japanese patient with HCC Highly divergent from others human HBV strains Likely a genotype C–gibbon *O**rthohepadnavirus* recombinant strains

Data was taken from Kramvis, 2014 [21], McNaughton et al., 2019 [5], and Schaefer, 2007 [22]. Abbreviations: HBV, hepatitis B virus; bp, base pair; aa, amino acids; G, guanine; A, adenine; T, thymine; BCP, basal core promoter; HCC, hepatocellular carcinoma.

**Table 2 viruses-13-01167-t002:** Viral and serological markers; and NK and T cells responses during chronic HBV stages.

Phase	I	II	III	IV	V
	HBeAg (+) Chronic HBV Infection	HBeAg (+) Chronic Hepatitis B	HBV Chronic Infection HBeAg (−)	Chronic Hepatitis B HBeAg (−)	HBsAg (−)
	HBeAg (+) Non-Inflammatory	HBeAg (+) Immune Active Phase	Inactive Carrier	HBeAg (−) Immune Active Phase	Occult Infection
Previous term	Immune tolerant	Immune reactive HBeAg (+)	Inactive carrier	HBeAg (−) chronic hepatitis	Occult infection
HBeAg	+++	++	−	−	−
Anti-HBe	-	+	+	+	+ or −
HBsAg	++++	+++	++ or + Few patients develop spontaneous clearance (~1% annual)	++	−
Anti-HBs	-	+++	++	+ or −	+ or −
HBV-DNA (IU/mL)	10^7^	10^4^–10^7^	<2 × 10^3^ ** or undetectable	>2 × 10 ^3^	Very low levels in liver and/or serum
ALT (IU/mL)	Normal	Elevated	Normal (~40)	Persistently or intermittently elevated	Normal
Hepaticdisease	None/minimal necroinflammation or fibrosis	Moderate/severe liver necroinflammation and accelerated progression of fibrosis	None/minimal necroinflammation and low fibrosis	Moderate/severe necroinflammation and fibrosis	Immunosuppression could lead to viral reactivation. Low risk of cirrhosis or HCC
NK cells innate response	Impaired production of IFN-γ and TNF-α	Impaired production of IFN-γ and TNF-α Phenotypically activated (↑ NKp44 in CD56^bright^ and CD69 in CD56^dim^ NK cells)	Restored production of IFN-γ	Phenotypically activated (↑ NKp44 in CD56^bright^ and CD69 in CD56^dim^ NK cells)	
	Normal NK cells cytolytic activity without differences in CD107a, perforin, or granzyme B expression.				
Global and HBV specific T cell responses	↓ IFN-γ ^#^ and TNF-α	↓ IFN-γ ^#^ and TNF-α	↑ HBsAg and HBcAg response producing IFN-γ	↑↑ IFN-γ ^#^ and TNF-α	
Progress risk	0.37% to 3.3% annual progression risk				
Others	Very low rate of spontaneous HBeAg loss Highly contagious. More frequent and prolonged in carriers infected by perinatal transmission Low efficacy of currently available treatments	Variable outcome, but most of them progress to stage III	Low risk of developing HCC or cirrhosis. Loss of HBsAg and/or spontaneous seroconversion occurs in 1–3% of cases	A low proportion of spontaneous remission of the disease	

Parameters based on the EASL 2017 Clinical Practice Guidelines [82]. NK cell and T cells responses were obtained from PBMC measures reported by Wang et al. [83]. ** HBV-DNA can be between 2 × 10^3^ and 2 × 10^4^ IU/mL in patients without a signal of CHB. ^#^ This is observed in patients, nonetheless controversial results are reported in animal models. Abbreviations: HBV, hepatitis B virus; HBeAg, HBV e antigen; HBsAg, HBV surface antigen; ALT, alanine aminotransferase; HCC, hepatocellular carcinoma; CD, cluster of differentiation; NK, natural killer cell; IFN-γ, interferon gamma; TNF-α, tumor necrosis factor alpha; (+), positive; (−), negative; IU, international units; mL, milliliters; >, more than; <, less than; ~, approximately; %, percentage; ↑, an increase in the expression; ↓, a decrease in the expression.

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
