# Peer review of "Molecular Mechanisms during Hepatitis B Infection and the Effects of the Virus Variability"

_viruses, 2021, doi:10.3390/v13061167_

Round 1

Reviewer 1 Report

The manuscript entitled ‘Molecular Mechanisms During Hepatitis B Infection and the Effects of the Virus Variability’ describes mechanisms involved during HBV infection. Emphasis is on analyzing the immunological mechanisms that regulate the infection and elimination of the virus. Although the article deals with some important topics that should be of interest to workers in the field. Concerns that to address before publication are listed below:

  • Overall the manuscript is very detailed, which of course is in an attempt to make the article comprehensive. However, article is a difficult read and the logical sequence of what the content is sometimes blurred by the details.
  • Related to the above point, there seems to be repetition of some of the sections and the same topics are covered from different angles in different sections of the manuscript. An example is the topic of the innate immune response to HBV, which is alluded to in early sections, then addressed directly in section 10.
  • A feature of the HBV genome that has an influence on mutation rates is its compactness with overlapping open reading frames. Implications of this point should be discussed.
  • Infection by horizontal transmission in children is not discussed.
  • Transmission from mother to child is usually perinatal, not vertical. The virus rarely passes through the placenta and it is rather during natural birth (perinatal) that infection of babies occurs.
  • Several points raised on lines 236 to 246 do not cite supporting literature.
  • The conclusion is bland and does not provide an insightful analysis of the field. The authors should provide a carefully considered opinion about key focus areas, where the field is going and what needs to be achieved to make progress towards cure.
  • The abbreviation is RIG-I, not RIG-1. Check on correct use of the abbreviations for ACLF and ALF.
  • Throughout the manuscript there are many grammatical, spelling and typographical errors that need correction.
  • The esthetics of the Figures should be improved. As with the text, there is a lot of detail and distillation of the key points would make the Figures easier to interpret.

Author Response

The reviewers indicate some comments, and we answer as follow:

REVIEWER 1:

NOTE: We highlighted changes in the manuscript with the following code: Green, new writing to make it fluently; yellow, new information; and blue, replacement and edition.

  • Overall the manuscript is very detailed, which of course is in an attempt to make the article comprehensive. However, article is a difficult read and the logical sequence of what the content is sometimes blurred by the details.

Answer: We performed major changes in the sequence and in the content in order to make clear for the readers mainly in “hepatitis B infection” and “immunopathogenesis” sections. Moreover, the titles of some sections were adjusted to be more specific with the content of the information.

  • Related to the above point, there seems to be repetition of some of the sections and the same topics are covered from different angles in different sections of the manuscript. An example is the topic of the innate immune response to HBV, which is alluded to in early sections, then addressed directly in section 10.

Answer: All the content of the manuscript was carefully reviewed and the repetitive information was only covered in the most adequate section.

  • A feature of the HBV genome that has an influence on mutation rates is its compactness with overlapping open reading frames. Implications of this point should be discussed.

Answer: A paragraph in section 2 was added “evolution rate and diversity of the HBV”

  • Infection by horizontal transmission in children is not discussed.

Answer: In the section of natural history of HBV infection was added information related to this point.

  • Transmission from mother to child is usually perinatal, not vertical. The virus rarely passes through the placenta and it is rather during natural birth (perinatal) that infection of babies occurs.

Answer: This statement was corrected in the manuscript

  • Several points raised on lines 236 to 246 do not cite supporting literature.

Answer: This paragraph was performed by authors to introduce the information in the “Response in Acute hepatitis B infection”

  • The conclusion is bland and does not provide an insightful analysis of the field. The authors should provide a carefully considered opinion about key focus areas, where the field is going and what needs to be achieved to make progress towards cure.

Answer:   This section was modified

  • The abbreviation is RIG-I, not RIG-1. Check on correct use of the abbreviations for ACLF and ALF.

Answer: The abbreviation RIG-I was corrected and related to ACLF and ALF we made a carefully review in our manuscript to be sure that these abbreviations were used correctly in relation with the clinical stage referred.

  • Throughout the manuscript there are many grammatical, spelling and typographical errors that need correction.

Answer: The manuscript was reviewed by an expert in “academic writing” with background in medical sciences.

  • The esthetics of the Figures should be improved. As with the text, there is a lot of detail and distillation of the key points would make the Figures easier to interpret.

Answer: The figures were modified to improve the esthetics and content.

Reviewer 2 Report

Campos-Valdez et al. nicely overviewed current knowledge about the impact of viral variability. This is a well-written review article and helps us comprehensively understand HBV biology with front-line research. This reviewer has a couple of minor comments to clarify the description.

  1. One of the most important consequences of HBV is liver cancer development. That would be more comprehensive if the authors can add a small section describing the association between HCC development and HBV variance and their mechanism if possible.
  2. In figure 4, annotations for different cell types are shown only in a single cell for each. Cell type legends with cartoons on the right blank area would make it easy to identify which is which.
  3. In figure 5, which cells are the center? I guess Treg. I would suggest putting Treg cartoons at the beginning of lines. The current figure looks like Treg can affect only Th.

Author Response

The reviewers indicate some comments, and we answer as follow:

NOTE: We highlighted changes in the manuscript with the following code: Green, new writing to make it fluently; yellow, new information; and blue, replacement and edition.

RESPONSE TO REVIEWER 2:

  • One of the most important consequences of HBV is liver cancer development. That would be more comprehensive if the authors can add a small section describing the association between HCC development and HBV variance and their mechanism if possible.

Answer: A section about “HBV related hepatocellular carcinoma” was added to the manuscript.

  • In figure 4, annotations for different cell types are shown only in a single cell for each. Cell type legends with cartoons on the right blank area would make it easy to identify which is which.

Answer: Figueres were modified trying to improve the esthetics and figure 4 cell type with legends with cartoons were added on the right blank area. 

  • In figure 5, which cells are the center? I guess Treg. I would suggest putting Treg cartoons at the beginning of lines. The current figure looks like Treg can affect only Th.

Answer: Figure 5 was modified to make more emphasis in the Treg cell function on tolerance state in chronic hepatitis B.

Round 2

Reviewer 1 Report

The manuscript has been improved as a result of the revisions.

1  Minor spelling and grammar errors remain, and these should be corrected.

2  The conclusion is improved, however the authors do not provide an opinion about where the research priorities should be focused to enable the field to progress significantly.

Author Response

The manuscript has been improved as a result of the revisions.

  1. Minor spelling and grammar errors remain, and these should be corrected.

Answer: The manuscript was reviewed carefully and were corrected spelling and grammar errors.

  1. The conclusion is improved, however the authors do not provide an opinion about where the research priorities should be focused to enable the field to progress significantly.

Answer: The conclusion was modified highlighting research areas to progress significantly in the hepatitis B virus infection area.

Thank you very much for your commentaries and advices. We think the paper improve considerably by your recommendations.